# Preparatory attentional templates in prefrontal and sensory cortex encode target-associated information

**Zhiheng Zhou**[1,2]*, **Joy Geng**[2,3]*

[1]College of Psychology, Sichuan Normal University, Chengdu, China; [2]Center for Mind and Brain, University of California, Davis, Davis, United States; [3]Department of Psychology, University of California, Davis, Davis, United States

## eLife Assessment

This **important** study decoded target-associated information in prefrontal and sensory cortex during the preparatory period of a visual search task, suggesting a memory component of human subjects performing such visual attention task. The evidence supporting this claim is **compelling**, based on multivariate pattern analyses of fMRI data. The results will be of interest to psychologists and cognitive neuroscientists.

**Abstract** Visual search relies on the ability to use information about the target in working memory to guide attention and make target-match decisions. The 'attentional' or 'target' template is thought to be encoded within an inferior frontal junction (IFJ)-visual attentional network. While this template typically contains veridical target features, behavioral studies have shown that target-associated information, such as statistically co-occurring object pairs, can also guide attention. However, preparatory activation of associated information within the IFJ-visual attentional network has never been demonstrated. We used fMRI and multivariate pattern analysis to test if target-associated information is explicitly represented in advance of visual search. Participants learned four face-scene category pairings and then completed a cued visual search task for a face. Face information was decoded in the fusiform face area, superior parietal lobule, and dorsolateral prefrontal cortex during the cue period, but was absent during the delay period. In contrast, associated scene information was decoded in the ventrolateral prefrontal cortex during the cue period, and most importantly, in the IFJ and the parahippocampal place area during the delay period. These results are a novel demonstration of how target-associated information from memory can supplant the veridical target in the brain's 'target template' in anticipation of difficult visual search.

*For correspondence:
zhou@sicnu.edu.cn (ZZ);
jgeng@ucdavis.edu (JG)

**Competing interest:** The authors declare that no competing interests exist.

## Introduction

We engage in visual search repeatedly throughout the day whenever we look for something, such as a friend in a café, a phone on the table, or ingredients at a store. When visual search is efficient, behavior feels frictionless, but failures result in frustration or even the derailment of behavioral goals. The canonical mechanism supporting visual search is thought to involve a network of frontal regions that maintain target information in a working memory 'template' that is used to adjust gain in sensory neurons that encode matching features (*Chelazzi et al., 1993*; *Desimone and Duncan, 1995*; *Liu et al., 2007*; *O'Connor et al., 2002*; *O'Craven et al., 1999*; *Reynolds and Heeger, 2009*; *Serences and Boynton, 2007*; *Stokes et al., 2009*; *Sylvester et al., 2007*; *Treue and Martínez Trujillo, 1999*). Preparatory changes in sensory gain lead to increases in baseline activity that enhance subsequent

stimulus-evoked responses by target-matching stimuli (*Boettcher et al., 2020*; *Chelazzi et al., 1993*; *Desimone and Duncan, 1995*; *Gayet and Peelen, 2022*; *O'Connor et al., 2002*; *Peelen and Kastner, 2011*; *Serences and Boynton, 2007*; *Stokes et al., 2009*; *Sylvester et al., 2007*; *van Loon et al., 2018*; *Witkowski and Geng, 2023*). In this way, information in the target template is thought to dictate sensory processing priorities that impact the ability to perceive and act on behaviorally relevant objects. Despite its importance for behavior, the bulk of work on target templates in the brain has focused on target veridical features (e.g., the color 'red' or small round shapes when looking for an apple). Few studies have examined how non-target information might also be held in the template and used as a proxy for the target. The current study tests if objects that are statistically associated with the target are ever prioritized in the target template in conjunction with, or even instead of, the actual target in anticipation of visual search (*Boettcher et al., 2018*; *Helbing et al., 2022*; *Mack and Eckstein, 2011*; *Yu et al., 2023*; *Zhou and Geng, 2024*).

Information in the target template is considered a source of goal-directed sensory-motor control (*Nee, 2021*). Recently, the posterior end of the inferior frontal sulcus and its junction with the precentral sulcus has been identified as holding object and feature-based information consistent with the definition of target template (*Baldauf and Desimone, 2014*; *Bedini et al., 2023*; *Bichot et al., 2015*; *Gong et al., 2022*; *O'Reilly, 2010*; *Soyuhos and Baldauf, 2023*; *Witkowski and Geng, 2023*; *Zanto et al., 2010*). These regions include a posterior-to-anterior cortical organization labeled as premotor eye field (PEF), posterior inferior frontal junction (IFJp), and anterior inferior frontal junction (IFJa) in the Human Connectome Project Multi-Modal Parcellation (HCP-MMP1; *Bedini et al., 2023*; *Glasser et al., 2016*). For example, *Baldauf and Desimone, 2014* found that attention to superimposed faces versus houses resulted in induced gamma synchrony between the IFJ and fusiform face area (FFA) or parahippocampal place area (PPA), respectively; this was coupled with selective sensory enhancements in stimulus processing. A likely homologue of this region has also been identified in non-human primates in the ventral prearcuate region (VPA) of the prefrontal cortex (*Bichot et al., 2015*; *Bichot et al., 2019*). Consistent with its role as a source for feature-based attention, inactivation of VPA resulted in poorer visual search performance and reduced V4 visual responses to the target. These results provide evidence for a causal link between the target representation in IFJ and sensory biases in V4 that facilitate visual search behavior. Together, the data suggest that regions encompassed by IFJ maintain target information and provide feedback signals to visual neurons encoding those same features. The role IFJ plays for feature-based attention appears to be analogous to that of the frontal eye fields (FEFs) for spatial attention (*Bichot et al., 2015*; *Corbetta et al., 2008*; *Moore et al., 2003*; *Ruff et al., 2006*; *Thompson et al., 2005*).

In addition to IFJ, which appears to encode the source of target information, visual search involves a number of other cognitive control functions that recruit other frontal and parietal regions. In particular, the dorsolateral and ventrolateral prefrontal cortices (dLPFC and vLPFC) are involved in proactive and reactive cognitive control, including the maintenance and manipulation of goal-relevant information across stimulus types, and flexibly updating those goal-based representations when unexpected errors are encountered (*Badre and Nee, 2018*; *Bettencourt and Xu, 2016*; *Braver et al., 2009*; *Christophel et al., 2018*; *Christophel et al., 2017*; *Emrich et al., 2013*; *Ester et al., 2015*; *Finn et al., 2019*; *Lee et al., 2013*; *Liu et al., 2003*; *Long and Kuhl, 2018*; *Nee and D'Esposito, 2017*; *Soon et al., 2013*; *Stokes et al., 2013*). Working memory representations in lateral prefrontal and parietal regions are also engaged in cognitive control computations that are task non-specific but essential to task functioning (*Bettencourt and Xu, 2016*; *Kwak and Curtis, 2022*; *Long and Kuhl, 2018*; *Panichello and Buschman, 2021*). Thus, a network of frontoparietal areas is expected to share prospective encoding of target information, and we include these regions of interest (ROIs) in our analyses, defined by the HCP-MMP1 (*Glasser et al., 2016*) and the 17-network atlases (*Schaefer et al., 2018*; see 'Materials and methods).

The ability to flexibly update goals is an important aspect of visual search because finding a target requires an iterative cycle of choosing an object based on the current template to attend to or look at, and then deciding if it is the correct target or not (*Alexander and Zelinsky, 2011*; *Hout and Goldinger, 2014*; *Malcolm and Henderson, 2010*; *Yu et al., 2022*; *Yu et al., 2023*). When it is the target, search is terminated, but when it is not, attention is guided to a new potential target and the cycle continues. Thus, targets must be maintained in memory over time, while control mechanisms flexibly adjust goals based on sensory outcomes (*Badre and Wagner, 2007*; *Egner, 2023*; *Kurtin*

*et al., 2023*; *Poskanzer and Aly, 2023*; *Rossi et al., 2009*; *Tünnermann et al., 2021*). Target information in multiple ROIs may evolve over time for these different computations, but we expect IFJ to encode the target template prospectively to facilitate attentional guidance and target decisions.

In contrast to the relatively straightforward idea that veridical target features are held in working memory and used to bias sensory processing, there is substantial evidence that during naturalistic scene viewing, attention is often guided by objects that are associated with the target but not the target itself (*Boettcher et al., 2018*; *Helbing et al., 2022*; *Mack and Eckstein, 2011*; *Zhou and Geng, 2024*). For example, people are faster to find targets (e.g., sand toys) that are near associated 'anchor' objects (e.g., sandbox), because the anchor serves as a large and more easily perceived spatial predictor of the target's location (*Boettcher et al., 2018*; *Helbing et al., 2022*; *Turini and Võ, 2022*). Using such proxy information is beneficial because it reduces the space over which the target must be searched for (*Boettcher et al., 2018*; *Castelhano and Krzyś, 2020*; *Hall and Geng, 2024*; *Josephs et al., 2016*). A recent study found that target-associated anchor objects implied by a visual scene could be decoded in parietal and visual cortex before the target appeared (*Lerebourg et al., 2024*), supporting the notion that objects associated with the target are used to bias sensory processing and guide attention (*Gayet and Peelen, 2022*; *Peelen et al., 2024*; *Yu et al., 2023*). Despite these studies, it remains unknown if target-associated information in memory is encoded in attentional control networks during the preparatory period at all, and if it is, if it occurs along with target information or instead of target information.

In this study, we address this gap in knowledge by asking if associated information in memory can be found in the IFJ, frontoparietal attention network, and category-selective sensory cortex during the delay period in anticipation of visual search. Such an outcome would suggest that information associated with the target is prioritized in preparation for visual search, potentially even at the expense of the target itself. We base our paradigm on behavioral work showing that participants can learn new face-scene category associations and use this information to guide target search, but only when target-distractor discrimination is difficult (*Zhou and Geng, 2024*). In this previous study, we trained participants on four face-scene associations in advance of a cued-visual search task. During visual search, a single face was cued to indicate the target on each trial. Next, a search display appeared with two lateralized faces superimposed on scenes. The target appeared on its associated scene on 75% of 'scene-valid' trials. Different sub-experiments used a variety of scene-invalid trials, including ones in which the associated scene was never present, was co-located with the distractor only, or

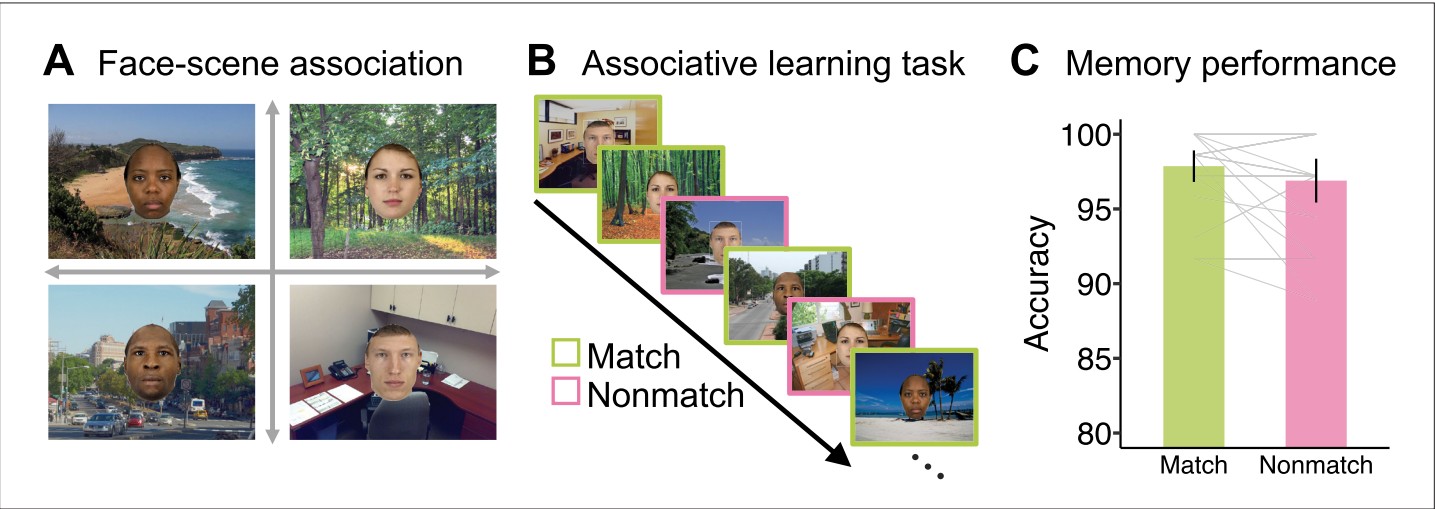

**Figure 1.** Associative learning task. (**A**) The four target face stimuli and their associated scenes. (**B**) Associative learning task to test memory for face-scene pairs. Participants viewed a series of face-scene pairs and made a judgment about whether the face and scene were matched or not. (**C**) Memory performance for both match and nonmatch conditions was high, suggesting a strong association was formed before the face search task. Error bars refer to 95% CIs. Facial images from The Chicago Face Database (*Ma et al., 2015*).

The online version of this article includes the following figure supplement(s) for figure 1:

**Figure supplement 1.** The four target face stimuli and their three distractor face counterparts.

**Figure supplement 2.** The four scene categories used in the search task.

appeared with both the target and the distractor. Here, we only use one of the 'scene-invalid' conditions because the search trials are not of primary interest. The main purpose of the study is to determine if the face cue activates a memory representation of the associated scene during the delay period in attentional control regions that encode the target template, namely IFJ. This outcome would be novel evidence for the hypothesis that the associated scene is represented within the target template during the delay period for the purpose of facilitating visual search. Such a result would be consistent with studies in humans and non-human primates showing that preparatory activation of target information in prefrontal and visual cortex is used to guide attention and make target decisions during visual search (*Baldauf and Desimone, 2014*; *Bichot et al., 2015*; *Chelazzi et al., 1993*; *Gregoriou et al., 2014*; *Kastner et al., 1999*; *Soyuhos and Baldauf, 2023*; *Witkowski and Geng, 2023*).

## Results

### Behavioral tasks

All participants (N=26) were first trained outside the scanner on four face-scene category associations (*Figure 1A*). First, they saw the displays introducing a face with a scene, for example, the text 'Emma is a lifeguard and can be found on the beach' presented with an image of a face, 'Emma', superimposed on a beach scene (see 'Materials and methods'). Second, participants were given a match-nonmatch decision task in which they indicated if a specific face-scene pair was a previously learned pair (*Figure 1B*). All participants reached a decision accuracy greater than 90%, with mean accuracy = 97.4% ± 2.7% (*Figure 1C*).

During the scan session, participants were given a cued visual search task (*Figure 2A*). Each trial began with a *search cue* of the target face for 1 second, followed by an 8-second blank *search delay* period, and then a search display for 0.25 second. Trials ended with an 8.75-second inter-trial interval to allow the BOLD response to return to baseline. The delay period was the temporal interval of greatest interest. While the cue-evoked stimulus response should contain decodable face information, any scene information during the delay period must reflect memory-evoked representations because no scene information was visually presented with the face cue. Search trials were composed of two face stimuli superimposed on scene images. The target appeared on the associated scene on 75% of trials (scene-valid) and on one of the three other scenes on 25% of trials (scene-invalid). Thus, the scene was probabilistically predictive of the target's location but not deterministic. The distractor face was a race-gender match. We previously found that this distractor condition was sufficiently difficult to elicit the use of scene information for guiding attention (*Zhou and Geng, 2024*). After the main search task, participants completed a face and scene 1-back task (*Figure 2B*). Data from this task were used as a benchmark test set for neural activation patterns trained in response to faces and scenes from the main visual search task.

### Cue-evoked face information is translated into a preparatory scene template prior to visual search

To test if face and/or scene information was present in the IFJ, frontoparietal attention networks, and category-selective visual cortex during the cue and delay periods prior to visual search, we adopted a cross-classification multivariate pattern analysis (MVPA) approach (*Figure 2C*). Separate classifiers were created for the cue and delay periods, but both classifiers were trained on data from the main search task and tested on data from the separate 1-back task using trials with the same face and scene stimuli. Analyses were focused on 12 ROIs within the frontoparietal network and the sensory cortex that are known to be involved in representing task structure, target templates for feature-based attention, and category-selective visual processing (*Figure 2D*). Frontoparietal ROIs were defined by the HCP-MMP1 (*Glasser et al., 2016*) and the 17-network atlases (*Schaefer et al., 2018*). Category-selective visual regions for faces (FFA) and scenes (PPA) in the ventral visual pathway were identified for each participant using the independent functional localizer task (*Stigliani et al., 2015*).

The cross-classification procedure for the cue stimulus involved training the classifier on activation patterns in response to the *search cue* in the face search task, and testing on activation patterns in response to *face* and *scene* stimulus *samples* seen during an independent 1-back task (*Figure 2C*). We expected the *search cue* classifier to successfully decode face stimuli but not necessarily scene stimuli since this time period should primarily reflect the stimulus-evoked sensory processing of the face cue.

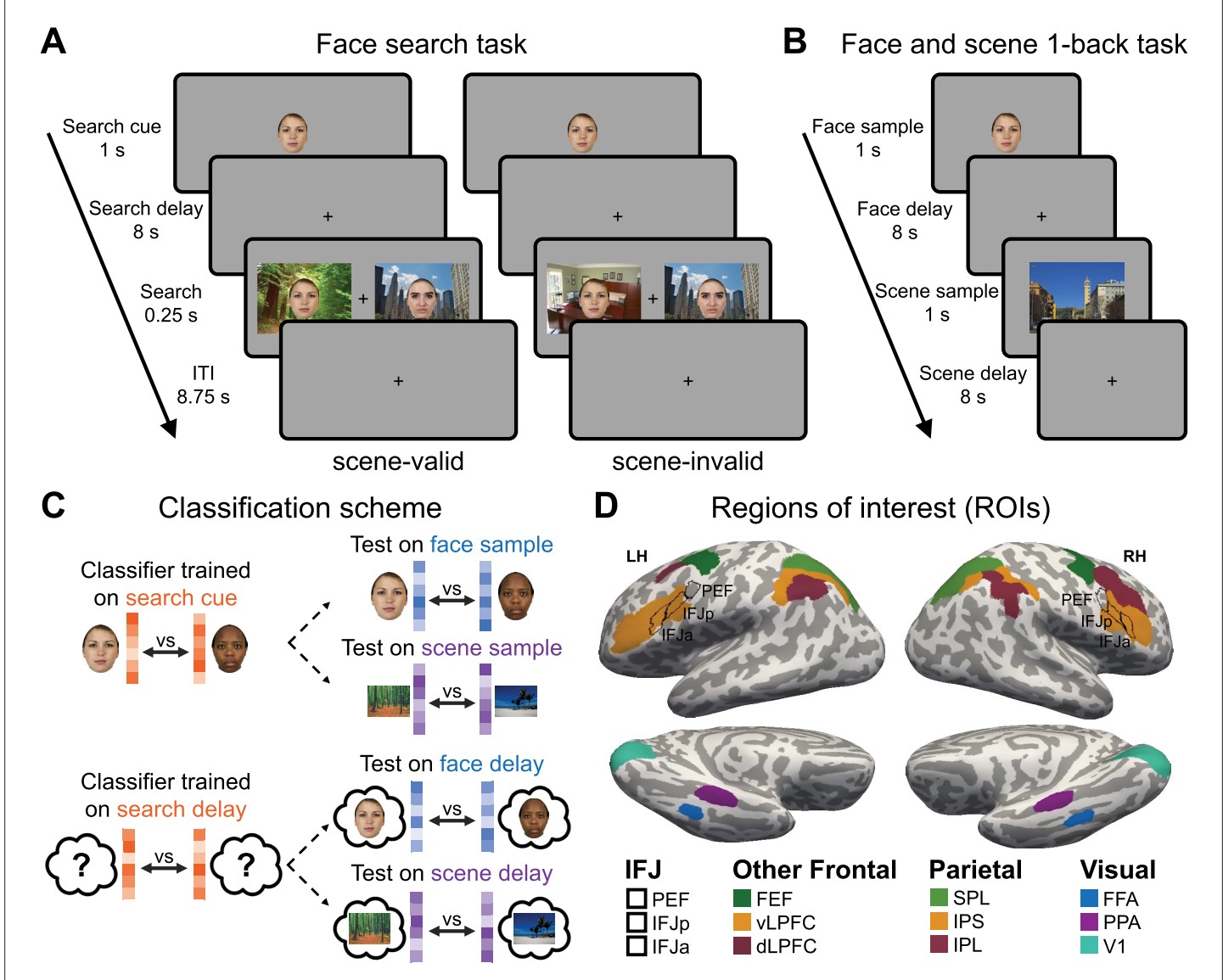

**Figure 2.** fMRI tasks and analysis procedure. (**A**) Face search task. Each trial started with a 1-second *search cue* indicating the target face for that trial. This was followed by an 8-second blank *search delay* period, and then the search display for 0.25 second. Participants pressed a button to indicate the location of the target on the left or right. (**B**) Illustration of the separate 1-back task used for cross-task classifier testing. Consistent with the main visual search task, the image was presented for 1 second followed by an 8-second *delay* period. (**C**) Classification scheme. Classifiers were trained on the neural response patterns from the *search cue* face stimulus and *delay* periods from the visual search task; the classifier was tested on face or scene *sample* stimulus and *delay* period from the 1-back task. (**D**) Visualization of the twelve functional regions of interest (ROIs) on the cortical surface of a representative participant. All ROIs were defined in an individual's native space. Facial images from The Chicago Face Database (*Ma et al., 2015*).

The online version of this article includes the following figure supplement(s) for figure 2:

**Figure supplement 1.** Illustration of overlap between the HCP-MMP1 atlas (*Glasser et al., 2016*) and the Schaefer resting-state 17-network atlas (*Schaefer et al., 2018*) in the inferior frontal junction regions.

The *delay* period cross-classification procedure involved training the classifier on patterns of activation during the 8-second *search delay* period after the face *search cue* had disappeared and testing the classifier on activation patterns from the 8-second *delay* following the *face* or *scene* sample stimulus from the 1-back task (*Figure 2C*). Here, we hypothesized that we would see decoding of scene information in the IFJ-PPA network either in conjunction with face information in IFJ, or even in place of it. Critically, the decodable information during the delay period for the *face* could be contaminated by the stimulus-driven activity during the cue. This is because the temporal separation between the

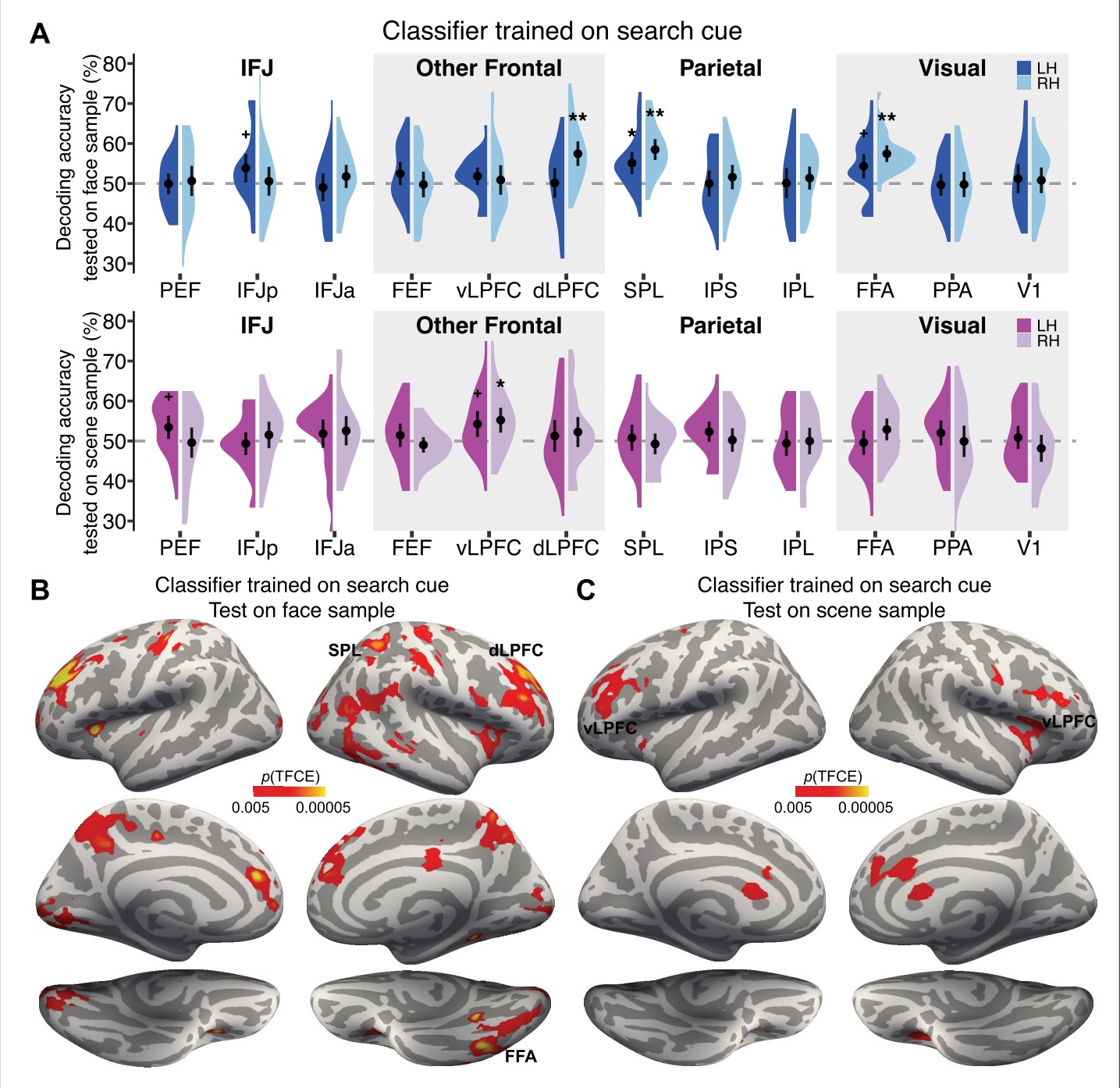

**Figure 3.** Decoding of face and scene information during the *search cue* period. (**A**) Evidence of face information in a priori defined regions of interest (ROIs). Greater than chance-level classification accuracies were found in the dorsolateral prefrontal cortices (dLPFC), superior parietal lobule (SPL), and fusiform face area (FFA). Evidence for scene information was found in ventrolateral prefrontal cortices (vLPFC). N=26 participants. †p<0.05 uncorrected, *p<0.05, **p<0.005. (**B, C**) Significant brain regions revealed by a whole-brain searchlight procedure with information about the face cue (**B**) or associated scene (**C**). Note that scene information was never shown during the cue period and therefore decoding of scene information reflects memory-evoked responses to the cued face.

face and the delay period is too short for the BOLD response to the face cue to completely resolve. However, this is not a concern for *scene* decoding because the scene is never visually shown during the cue period. Thus, the ability to decode scene information during the delay period has to reflect memory-evoked representations in response to seeing the specific face cue.

Application of this approach to the *search cue* period (**Figure 3A**) revealed significant decoding of face information in bilateral FFA (left p=0.0074, uncorrected; right p<0.005). As expected, each of the four face cues could be distinguished from each other in FFA. Faces were also significantly decoded in bilateral superior parietal lobule (SPL; left p=0.018; right p<0.005), right dLPFC (p<0.005), and left IFJp (p=0.012, uncorrected), consistent with the storage of face stimuli in working memory for task-based control. In contrast, scene information was only decoded in bilateral vLPFC (left p=0.0066, uncorrected; right p=0.023); there was no significant decoding of scene information in the scene-selective regions PPA (left p=0.12; right p=0.51). Decoding in vLPFC is important and consistent with the retrieval of associated-scene information from long-term memory in response to the face cue. In addition to the ROI results, we also conducted a whole-brain searchlight using a cluster-corrected threshold of *p*TFCE<0.005 to identify additional areas that may have encoded face or scene information during the *search cue* period. This revealed additional significant clusters where the face cue could be decoded in the left dLPFC (**Figure 3B** and **Supplementary file 1**). These results indicate that the *search cue* period is dominated by face information in a widespread network of sensory and frontoparietal regions, with limited scene information in vLPFC, related to the cognitive control of memory (**Badre and Wagner, 2007**).

In contrast to the *search cue* period, face information during the *search delay* period (**Figure 4A**) could only be decoded in the left intraparietal sulcus (IPS; p=0.019, uncorrected), perhaps reflecting working memory storage of the target face (**Bettencourt and Xu, 2016**; **Jeong and Xu, 2016**). However, the associated scene was now reliably decoded in bilateral PPA (left p=0.007; right p<0.005), bilateral PEF (left p=0.03; right p<0.005), and right IFJp (p=0.027, uncorrected). Additional searchlight analyses revealed clusters (*p*TFCE<0.05) in bilateral retrosplenial cortex (**Figure 4B** and **Supplementary file 2**). The results show robust decoding of scene information in the IFJ-PPA network only during the delay period, suggesting that the face cue was translated into a 'guiding template' of the associated scene in anticipation of visual search. Strikingly, the overall pattern shows a double dissociation between face information in both the prefrontal and category-selective visual cortex during the *search cue* period and scene information during the *search delay* period. Importantly, during the *search delay* period, the target face was not represented in the frontal attentional control regions, nor category-selective visual cortex. Whole-brain analyses found that no brain regions showed decoding of both face and scene information. Instead, the associated scene was activated during the delay period in IFJ and category-selective visual cortex, providing strong evidence for the hypothesis that target-associated information is activated from memory in preparation for guiding attention during target search. A final exploratory analysis was conducted by splitting the delay period into first and second half time periods. The results revealed no differences in the decoding of target-associated information in the first versus second half of the delay period (**Figure 4—figure supplement 1**).

## Scene-invalid trials require more cognitive control and delay target localization

The decoding results from the *search delay* period suggest that scene information is prioritized over that of the target face in preparation for visual search. If true, then scene-invalid trials should produce a prediction error and require reactive cognitive control to switch the search template back to the target face. The behavioral results support this interpretation: participants were more accurate (*t*(25) = 4.96, p<0.001, *d*=0.97) and had shorter RTs (*t*(25) = –3.46, p=0.002, *d*=0.68) on scene-valid compared to scene-invalid trials (**Figure 5A**).

To test for prediction error and reactive cognitive control, we conducted a univariate analysis contrasting scene-invalid and scene-valid trials. This 'validity effect' contrast showed greater activation on scene-invalid trials in bilateral IFJ extending to the middle frontal gyrus, bilateral IPS, insula, and anterior cingulate cortex (ACC; **Figure 5B**, **Figure 5—figure supplement 1** and **Supplementary file 3**). These regions are similar to those reported in previous studies when cognitive control is needed for task switching, attentional shifting, and resolving cognitive conflict (**Corbetta et al., 2008**; **Dombert et al., 2016**; **Guerin et al., 2012**; **Liu et al., 2003**). In particular, the combined activation of the lateral prefrontal cortex and ACC is indicative of conflict detection in ACC and the updating of goals in the lateral prefrontal cortex necessary for successful goal adaptation (**Egner, 2023**).

Moreover, looking specifically within our a priori ROIs, we find significant activation in bilateral IFJp, IFJa, vLPFC, and the left dLPFC and IPS during scene-invalid compared to scene-valid trials

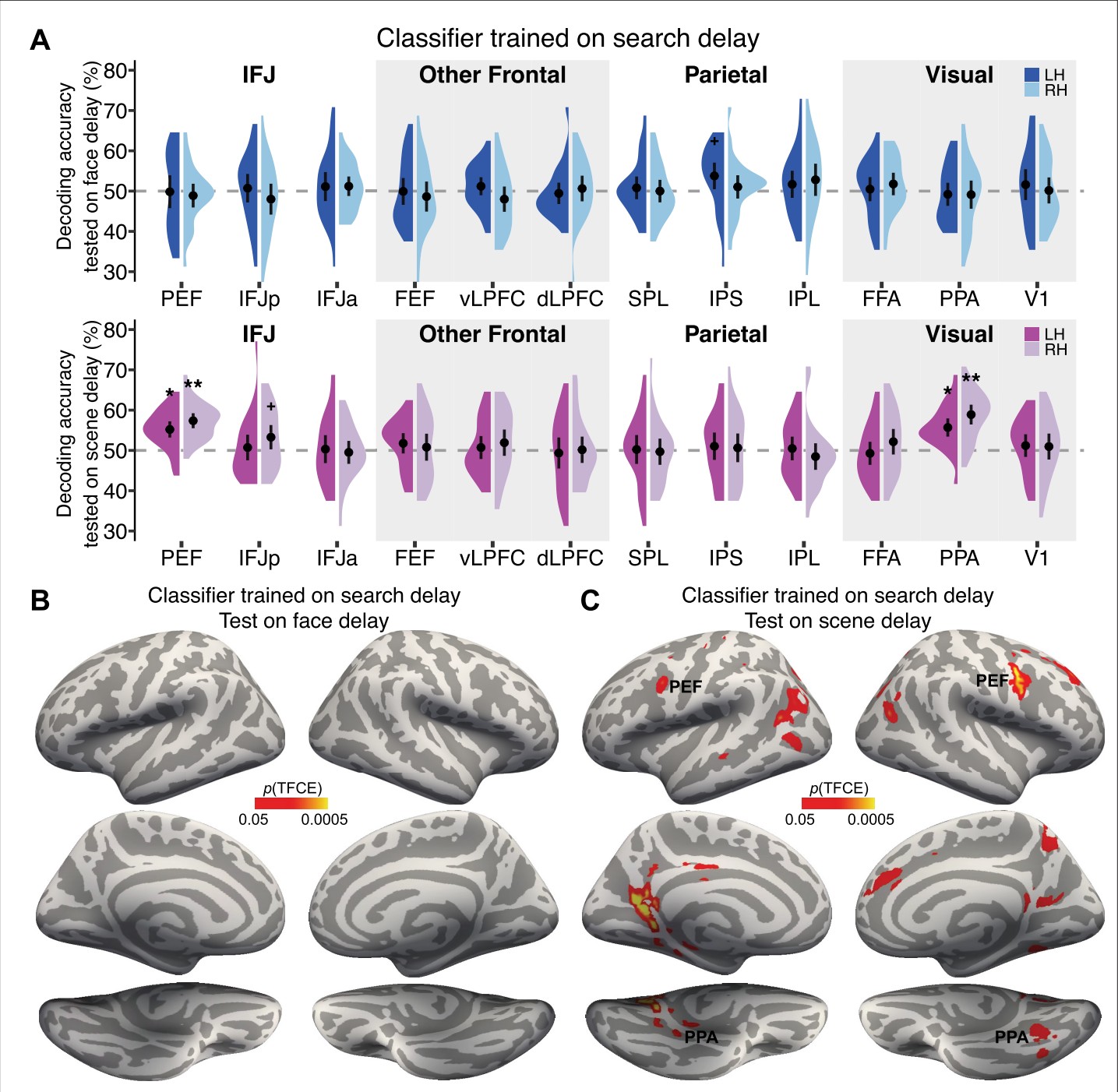

**Figure 4.** Decoding of face and scene information during the *search delay* period. (**A**) Evidence of face information in a priori defined regions of interest (ROIs) was only found in the left intraparietal sulcus (IPS). However, scene information was decoded in both inferior frontal junction (IFJ) and parahippocampal place area (PPA), reflecting memory-evoked target-associated information in a network that encodes the target template. N=26 participants. †p<0.05 uncorrected, *p<0.05, **p<0.005. (**B, C**) Whole-brain searchlight analyses showed no additional brain regions carried significant information about the face (**B**), but additional scene information was found in the retrosplenial cortex (**C**).

The online version of this article includes the following figure supplement(s) for figure 4:

**Figure supplement 1.** Decoding of face and scene information during the *search cue* (**A**), *search delay1* (**B**), *and search delay2* (**C**) periods.

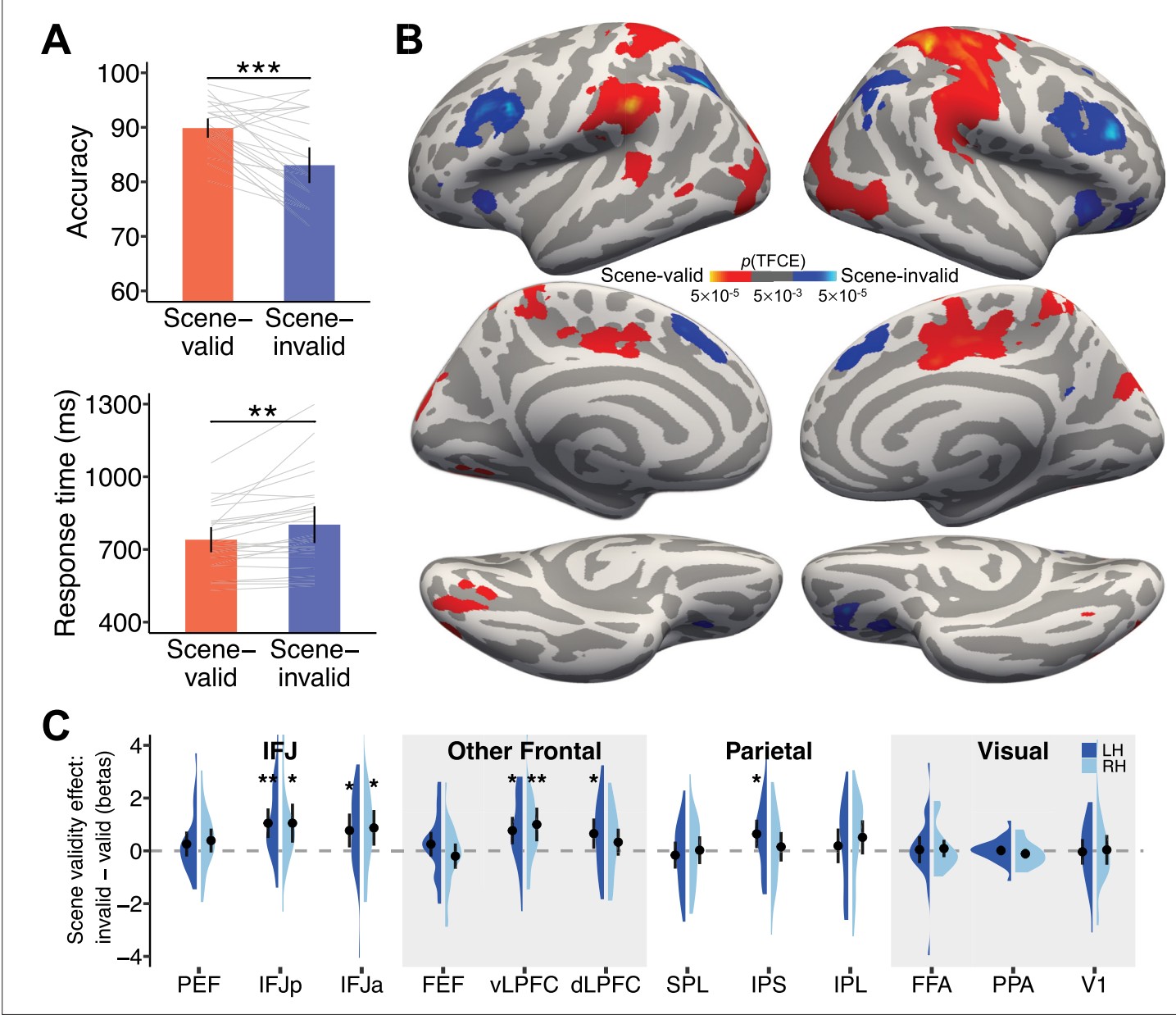

**Figure 5.** Behavioral and brain results from the face *search* period. (**A**) Behavioral accuracy and RT both showed a scene-validity effect, suggesting scene information was used to guide attention during search. N=26 participants. **p<0.005, ***p<0.001. Error bars refer to 95% CIs. (**B**) Whole-brain group-level univariate contrast results showing significantly greater activations for the scene-invalid than scene-valid conditions are illustrated in blue (cold colors), and the reverse contrast in red (hot colors). (**C**) Contrast betas from the scene-invalid minus scene-valid conditions within each of the a priori regions of interest (ROIs). *p<0.05, **p<0.005.

The online version of this article includes the following figure supplement(s) for figure 5:

**Figure supplement 1.** Univariate contrast results from the search period shown in a volumetric MNI standard brain.

(*Figure 5C*). Notably, scene-valid trials did not produce significantly greater activation in any of the ROIs, suggesting that scene-invalid trials required cognitive control to reset attentional priorities when expectations that the scene could be used to find the target were violated. Together, these results are consistent with the idea that scene-invalid trials required the active reinstatement of the target face into working memory and the target template to make a response. These results are consistent with scene-invalid trials producing a prediction error that requires flexible, reactive, cognitive control mechanisms to switch search templates (*Cole et al., 2013*), a process that slows responses and

reduces accuracy. Finally, it is worth noting that both the univariate contrast maps and ROI analyses revealed a potential distinction between the posterior IFJ subregion, PEF, and the relatively anterior IFJp and IFJa in feature-based attention and cognitive control (*Muhle-Karbe et al., 2016*; *Soyuhos and Baldauf, 2023*). Consistent with the definition of ROIs based on the two atlases, the IFJp/IFJa from the HCP-MMP1 overlaps with the vLPFC parcel from the 17-network, but the PEF does not overlap with the vLPFC (*Figure 2D*).

## Discussion

The canonical mechanism for visual attention involves preparatory changes in sensory gain that increases stimulus-driven responsivity when a matching stimulus appears (*Chelazzi et al., 1993*; *Corbetta et al., 2008*; *Kastner et al., 1999*; *Luck et al., 1997*; *Reynolds and Heeger, 2009*; *Serences and Boynton, 2007*; *Vossel et al., 2014*). This sensory modulation is thought to have its source in a network of frontoparietal regions that encode and hold target information as a 'template' for sensory modulation (*Chelazzi et al., 1993*; *Ester et al., 2016*; *Esterman and Yantis, 2010*; *Gong et al., 2022*; *Liu et al., 2003*; *Peelen and Kastner, 2011*; *Stokes et al., 2009*; *van Loon et al., 2018*). These studies provide a foundation for understanding how working memory representations that are distributed across frontal and parietal regions are used to bias visual processing towards relevant information for achieving behavioral goals. However, most of the work on this topic has focused on target-veridical features. Much less is known about how these mechanisms support adaptive updates to information used to find the target during active visual search.

In contrast to the neural evidence for veridical templates, there is considerable behavioral evidence showing that preparatory attention is flexible, highly adaptive, and sensitive to changing contexts. Many behavioral studies have shown that scene structure, statistically co-occurring object pairs, and large, stable, predictive 'anchor objects' are all used to guide attention and locate smaller objects (*Battistoni et al., 2017*; *Boettcher et al., 2018*; *Castelhano and Krzyś, 2020*; *Castelhano et al., 2009*; *Collegio et al., 2019*; *de Lange et al., 2018*; *Gayet and Peelen, 2022*; *Hall and Geng, 2024*; *Helbing et al., 2022*; *Josephs et al., 2016*; *Mack and Eckstein, 2011*; *Malcolm and Shomstein, 2015*; *Nah and Geng, 2022*; *Peelen et al., 2024*; *Võ et al., 2019*; *Yu et al., 2023*; *Zhou and Geng, 2024*). For example, in a previous behavioral study, we showed that when the target is hard to find, scene information is used as a proxy in the target template to guide attention toward the likely target location more efficiently (*Zhou and Geng, 2024*).

Despite the vast behavioral evidence for the use of non-target information in target-guided visual search, there has been little evidence so far for the presence of target-associated information in preparatory 'target templates' encoded in frontal-parietal-sensory attentional control networks. One exception is a recent study that found preparatory activation in the lateral occipital complex (LOC) for a target-associated anchor object in advance of target detection (*Lerebourg et al., 2024*). In this study, participants were asked to find either a book or a bowl located on two visually unique tables within two living room scenes. The scene context predicted which table would hold each object. When participants were given a target cue and shown a scene with the table regions masked, the associated table, but not the target, could be decoded from LOC. This study provided compelling evidence that target-associated information is preferentially represented in advance of object detection. However, because the tables were implied by the scene preview, it is somewhat unclear if decoding of the tables reflected the preparatory reinstatement of a target-associated memory or priming of task-relevant information by the scene. There was also no information about template source regions in the frontal cortex, such as the IFJ (*Baldauf and Desimone, 2014*; *Bedini et al., 2023*; *Soyuhos and Baldauf, 2023*).

Our study builds on this prior literature by teaching participants novel face-scene associations and then using the faces as cues and targets in a visual search task. Scenes were technically task-irrelevant in the search task and were not shown during the cue period. Thus, the presence of any pre-search preparatory scene information in the brain has to be reinstated from memory based on a previously learned association without direct visual prompting. This is important because any scene information represents memory-evoked response to a specific visually presented face cue. First, however, we confirmed that the identity of the face shown as the cue was decoded in bilateral FFA, right dLPFC, bilateral SPL, and left IFJp using a priori ROIs. Decoding in left dLPFC was found in subsequent searchlight analyses, which revealed symmetric clusters of voxels located bilaterally in the middle

frontal gyrus extending to the superior frontal gyrus (*Figure 3B* and *Supplementary file 1*). Overall, these results were as expected and showed that the onset of the face cue was represented in sensory and task-based control regions (*Chiu et al., 2011*; *Contreras et al., 2013*; *Guntupalli et al., 2017*; *Jeong and Xu, 2016*; *Long and Kuhl, 2018*; *Nestor et al., 2011*).

Next, and in contrast to the cue-evoked face representations, the only brain region containing scene category information during the cue period was in vLPFC. In prior studies, scene decoding has been found in vLPFC during perceptual scene categorization tasks (*Jung et al., 2018*; *Jung and Walther, 2021*; *Qin et al., 2011*; *Walther et al., 2009*), but more generally, vLPFC is engaged when cognitive control is required to adjudicate task-based memories (*Badre and Wagner, 2007*; *Egner, 2023*; *Levy and Wagner, 2011*; *Muhle-Karbe et al., 2018*; *Nee and D'Esposito, 2016*; *Neubert et al., 2014*; *Tamber-Rosenau et al., 2018*). This suggests that the face cue evoked the retrieval of scene category information from memory. The most important question is, therefore, whether this cue-related scene information in vLPFC would be translated into a preparatory search template.

We predicted that, if the associated scene was used as the target template, we should observe successful decoding of scene categories in scene-selective sensory cortex during the delay period, reflecting enhancements in sensory gain, and in parietal and lateral prefrontal cortex, reflecting scene information in working memory and cognitive control. It is well-documented that template information can be decoded in distributed cortical regions, but that each region likely plays a different computational role (*Ester et al., 2015*; *Ester et al., 2016*; *Gong et al., 2022*; *Lee and Geng, 2017*; *Long and Kuhl, 2018*; *Peelen and Kastner, 2011*; *Soon et al., 2013*; *Stokes et al., 2009*; *van Loon et al., 2018*; *Witkowski and Geng, 2023*). Consistent with this, we found that the associated scene category was decoded in IFJ regions and PPA during the delay period. Scene information decoded in vLPFC during the cue period appeared to trigger retrieval of the associated scene, which was turned into a scene template in IFJ and PPA in preparation for target search (*Badre and Wagner, 2007*; *Xu et al., 2022*). Additional searchlight analyses revealed clusters in bilateral retrosplenial cortex that contained scene information (*Figure 4C* and *Supplementary file 2*).

Interestingly, no face information was decodable from FFA or IFJ during the search delay period, suggesting it was not actively maintained in the target template. However, there was evidence of decoding in IPS, indicating that face information was still held in working memory (*Bettencourt and Xu, 2016*; *Christophel et al., 2018*; *Ester et al., 2015*; *Favila et al., 2018*; *Gong et al., 2022*; *Jeong and Xu, 2016*; *Kwak and Curtis, 2022*; *Olmos-Solis et al., 2021*; *Yu and Postle, 2021*). It is also worth noting that there was no significant decoding of the face or scene information in FEF in the delay period, a region commonly found in top-down attentional control. This may be because there was no spatial prediction given by the cue in our task, as has been suggested by others (*Bedini and Baldauf, 2021*; *Bichot et al., 2015*).

The consequences of prioritizing associated scenes over the target face were seen in longer RT and more errors on scene-invalid trials. The univariate results suggested that scene-invalid trials triggered a prediction error in ACC that led to an updating of what information should be used to find the target in vLPFC (including IFJp/IFJa) and IPS. However, because activation was more widespread in this analysis, it was not possible to determine if the patterns of activation differed in subregions of IFJ and other areas of vLPFC. This pattern of data supports the notion that preparatory scene activity in the delay period supplanted the target in the 'target' template in service of facilitating behavior; when it was inaccurate, however, reactive cognitive control was necessary to update search adaptively (*Botvinick et al., 2004*; *Braver, 2012*; *Egner, 2023*; *Kolling et al., 2016*; *O'Reilly et al., 2013*). Our current findings are likely due to the utility of scene information for finding the face. In this task, the target and distractor faces were perceptually similar, but scene information was highly distinctive. If the target face was easy to discriminate, the utility of the scene disappears (*Zhou and Geng, 2024*). Under those conditions, we would not expect to see scene information loaded into the target template. Thus, we expect our results to generalize only when target-associated information is easier to discriminate than target information. Despite our clear evidence for scene activation in IFJ and PPA during the delay period and changes in visual search performance based on scene associations, it is not possible to derive causality between the two given the correlational nature of fMRI and behavioral performance.

In summary, our study builds on a growing body of literature on preparatory attention, focusing specifically on what information is used to generate the most predictive target template. We provide novel evidence that target-associated information can supplant the actual target in the 'target'

template when the associated information is expected to more easily facilitate behavioral goals. In our case, the cued target face was translated into a template for a probabilistically predictive associated scene in the IFJ and sensory cortex in preparation for search. This suggests that when there is uncertainty in target detection and decisions, additional information is dynamically recalled to facilitate performance (*Hansen et al., 2012*; *Summerfield et al., 2011*; *Witkowski and Geng, 2023*). The results are consistent with decades of behavioral work showing the adaptive nature of attention in supporting goal-directed behavior. It goes further in providing a first demonstration of how this mechanism unfolds using target-associated information from memory in frontal-parietal-sensory networks to anticipate visual search.

## Materials and methods

### Participants

The sample size was chosen based on a power analysis of a previous behavioral study (*Zhou and Geng, 2024*) and similar fMRI MVPA studies (*Albers et al., 2013*; *van Loon et al., 2018*). Using the more conservative estimate based on the behavioral validity effect size ($dz$ = 0.678), the estimated minimum sample size to achieve significant effects (p=0.05, two-tailed) with a power 0.9 was N=25. A final number of 26 participants (mean age = 21.1 years, range = 18–29 years, females = 14, males = 12) were recruited from the University of California, Davis, and were given a combination of course credit and monetary compensation ($50) for a 2-hour MRI session. All participants were right-handed and native English speakers or had spoken English before the age of 5. None of them reported a history of neurological or psychiatric disorders. In addition, they all had normal or corrected-to-normal vision and passed the color blindness test (https://colormax.org/color-blind-test/). Two additional participants were removed due to excessive motion during scanning (head movement >3 mm), and a third participant was removed due to falling asleep in the scanner. The study adhered to the ethical principles of the Declaration of Helsinki and the NIH guidelines for ethical research. All consent forms and procedures were approved by the Institutional Review Board of the University of California, Davis (IRB number 1296375).

### Stimuli

Stimuli for all experiments consisted of 16 faces (*Figure 1A*, *Figure 1—figure supplement 1*) selected from the Chicago Face Database (CFD, *Ma et al., 2015*) and 64 scenes (*Figure 1A*, *Figure 1—figure supplement 2*) selected from a scene categorization study (*Jung et al., 2018*). The ratings for gender and race for all faces were restricted to be larger than 0.9 (based on normalized rating scores ranging between 0 and 1 provided by the CFD; a higher score indicates being more representative of that category). As in our previous study (*Zhou and Geng, 2024*), we used four target faces derived from crossings between two genders (woman and man) and races (black and white). An additional three faces were selected from each category to serve as distractors for the study. An oval mask was used to crop all selected faces to reduce the visibility of extra-facial features, for example, hair. The scene categories were crossed between content (nature and urban) and layout (open and closed). There were 16 exemplar images from each scene category used in the experiment. The resolution of the face images was 150×200 pixels (2°×2.7° visual angles) and the scenes were 800×600 pixels (10.7°×8° visual angles). All stimuli were presented against a gray background (RGB: 128, 128, 128). The low-level luminance and contrast of all face and scene images were controlled with the SHINE_color toolbox (*Willenbockel et al., 2010*).

### Experimental design

#### Face-scene associative learning task

Before the MRI scan session, participants first learned four face-scene category associations (*Figure 1A*). During the learning phase, each of the face-scene pairs was shown along with a narrative, for example, "This is Holly. She is a ranger, so you will find her in the forest." Next, participants were tested for their memory on the newly learned associations in a face-scene pair matching task (*Figure 1B*). On each test trial, both a face and scene appeared on the screen, and participants were asked to judge whether the face was associated with the scene by pressing the 'm' for a match, or 'n' for a nonmatch. Feedback was given with the word 'correct' or 'incorrect' presented for 0.5 seconds

after each response. The inter-trial interval (ITI) was 0.5 seconds. Each of the four faces was tested on 18 scene-match trials and 9 scene-nonmatch trials. There was a total of 108 trials presented in a pseudorandomized order. For the 18 match trials, participants viewed the target face on 18 different scene exemplars from the same category. For the nine nonmatch trials, participants viewed the target face on three different scene images (randomly selected) from the remaining three scene categories.

### Face search task

After the learning task, participants completed the face search task in the first scan session (*Figure 2A*). At the beginning of each trial, a face cue was presented for 1 second followed by an 8-second fixation cross. Next, the search display appeared for 0.25 second. The search display was followed by an 8.75-second ITI. The search display was composed of the target face and one randomly selected distractor face. Each face was superimposed on a scene image. The face-scene pairs appeared on the left and right of the screen split by a central fixation cross. The faces were centered in each of the scenes and the distance from the inner edge of the scene to the central fixation was 1° visual angle. The target face was always presented while the distractor face was randomly drawn from the set of three faces that matched the target in both gender and race. Participants indicated the location of the target face by pressing the '1' or '2' key on the MRI-compatible keypad corresponding to the left or right side. The response was limited to within 2 seconds.

Search trials were divided into scene-valid and scene-invalid conditions. The scene was valid on 75% of trials and invalid on the remaining 25%. On the scene-valid trials, the target face was always superimposed on its associated scene while the distractor face was superimposed on a scene from one of the three unassociated categories. On scene-invalid trials, the scenes that appeared with the target and distractor faces were from two different scene categories, both of which were unassociated with the target face.

All participants completed eight runs of the face search task. There were 16 trials (12 scene-valid trials and 4 scene-invalid trials) in each run and 4 trials with each of the four target faces. The participants were not informed about the exact probability of the valid to invalid trials in the experiment. The trial order as well as the location of the target face in the search display was counterbalanced and pseudorandomized. Participants completed a full practice run in the scanner at the beginning to familiarize them with the task and the testing environment.

### Face and scene 1-back task

After completing the face search task, a 6-minute T1-weighted structural scan was collected. Then, each participant completed a 1-back task in a second scan session with four runs (*Figure 2B*). On each trial, one stimulus was presented at the center for 1 second, followed by an 8-second fixation. Participants were instructed to make a button press when the current stimulus was the same as the previous one. There were four trials for each target face and scene category. Different exemplars were used for each repetition of the same scene category. An additional six trials were designated as the 1-back trial with a repeated stimulus. The trial sequence was counterbalanced and pseudorandomized. Participants completed a full practice run in the scanner before the formal test.

### Functional category-selective localizer

We identified two ROIs, the FFA and the PPA, for each individual participant in an independent functional localizer task (*Stigliani et al., 2015*). Stimuli were 2-D grayscale images (~16°×16° for all stimuli) consisting of faces, houses, corridors, cars, instruments, and phase-scrambled images. Eight images from the same object category were on for 0.4 second and off for 0.1 second in a 4-second mini-block. Participants performed a 1-back memory task and repeats occurred in half of the blocks in each run. Each run consisted of 54 blocks evenly divided by the six stimulus categories, and the block order was counterbalanced and pseudorandomized across runs. Two runs were collected for each participant.

## Stimulus apparatus

The *Face-scene associative learning task* was conducted on a 14-inch MacBook Pro laptop with a spatial resolution of 1920×1080, and all stimuli were presented using the Testable platform (https://www.testable.org/). The remaining experimental tasks were conducted while the participants were in the scanner. All stimuli for the scanner experiments were generated using Psychotoolbox-3 installed

on a Dell desktop PC and displayed on a 24-inch BOLDscreen LCD monitor with a spatial resolution of 1920×1200 pixels. Participants viewed stimuli through a mirror attached to the head coil which projected the monitor ~120 cm away outside of the scanner bore.

## MRI acquisition and preprocessing

All scans were performed on a 3-Tesla Siemens Skyra scanner with a 32-channel phased-array head coil at the University of California, Davis. Functional scans using T2-weighted echoplanar imaging (EPI) sequence with an acceleration factor of 2 were acquired with whole-brain volumes of 48 axial slices of 3 mm thickness (TR/TE 1500/24.6 ms, flip angle 75°, base/phase resolution 70/100, 3×3 mm$^2$, FOV 210 mm). High-resolution structural MPRAGE T1-weighted images (TR/TE 1800/2.97 ms, flip angle 7°, base/phase resolution 256/100, FOV 256 mm, 208 sagittal slices) were acquired and were used for anatomical normalization, co-registration, and cortical surface reconstruction. The whole MRI session was finished within 2 hours.

Functional and structural data were preprocessed using SPM12 (Wellcome Department of Imaging Neuroscience), FreeSurfer (*Dale et al., 1999*; *Fischl et al., 1999*), and in-house MATLAB (MathWorks) code. The first six initial functional scans of each run were discarded to allow for equilibrium effects. The preprocessing for functional EPI data included slice-time correction and spatial realignment. Using a two-pass procedure, fMRI data from all three scan sessions were aligned to the mean of the EPI images of the first session run. Participants with head motion >3 mm were excluded. The structural image was coregistered with the mean image. Cortical hemispheric surface reconstruction was performed using the 'recon-all' command in the FreeSurfer. All fMRI analyses were performed in the native individual space without spatial smoothing, except that a 4 mm full-width at half-maximum (FWHM) smoothing was applied to the *functional category-selective localizer* runs.

The segmentation utility in SPM12 was applied to estimate gray and white matter boundary parameters for spatial normalization. Then, the analyzed native space functional data (i.e., univariate beta images and searchlight classification accuracy images) was normalized into the standard 2×2×2 mm$^3$ MNI reference space and smoothed with an 8 mm FWHM isotropic kernel for group-based whole-brain analyses.

## fMRI general linear model (GLM)

For each scan session, a GLM was generated by convolving the canonical hemodynamic response functions with experimental conditions for each voxel. Six motion realignment parameters in each experimental run were included as nuisance regressors to control for head motion confounds. For the *face search task*, the GLM was based on different time segments of the task corresponding to the face *search cue*, *search delay* period, and search display. The GLM was constructed with 11 regressors for each experimental run. The first four regressors had a duration of 1 second and were used to estimate the four different face search cue identities. The next four regressors were 8 seconds long and estimated the search delay period following each face. Three additional regressors with a duration of 1 second were used to model the three types of search displays: scene-valid trials, scene-invalid trials, and error response trials.

A separate GLM was used to model the *face and scene 1-back task*. The GLM was constructed with 17 regressors for each experimental condition in each run. There were eight regressors for each of the four face images and its corresponding delay period, and another eight regressors for each scene category image and its corresponding delay period. The BOLD responses to the stimulus image and its corresponding delay period were modeled separately, just as it was for the *face search task*. The same model parameters were used so that the two tasks would be compatible for our cross-task decoding procedure. The last regressor with 1-second duration was used to define the 1-back event, that is, response to a repeated image, which was not a trial type of interest.

A third GLM was used to model the *functional category-selective localizer* in order to identify FFA and PPA in each participant. The GLM was constructed with six regressors with a 4-second duration to estimate all object categories (faces, houses, corridors, cars, instruments, and phase-scrambled images) within two runs. Contrast beta images between specific types of stimulus categories were used to identify category-selective regions in the visual cortex at the individual level.

## ROIs definition

The ROIs of primary interest focused on subregions of the frontal, parietal, and visual cortex that are known to be involved in attentional and cognitive control. As shown in *Figure 2D and A*, a total of 12 ROIs were defined in the native space of each participant through a combination of the Human Connectome Project Multi-Modal Parcellation (HCP-MMP1; *Glasser et al., 2016*), the resting-state 17-network (*Schaefer et al., 2018*), and our independent category-selective localizer. Each ROI selected from the surface-based HCP-MMP1 atlas and the resting-state 17-network was first transformed from the group-based fsaverage surface into an individual surface, and then remapped onto the volumetric native space using the 'mri_surf2vol'.

The HCP-MMP1 atlas was used to define three ROIs corresponding to the IFJ (*Bedini et al., 2023*), located along the posterior-to-anterior axis of the inferior frontal sulcus, namely the premotor eyefield (PEF), posterior IFJ (IFJp), and anterior IFJ (IFJa). Using the resting-state 17-network, we extracted the dorsal attention network, the ventral attention network, and the frontoparietal control network ROIs from the lateral frontal and parietal cortex. The cortical parcellations in the three networks in the lateral prefrontal cortex correspond to the FEFs, vLPFC, and dLPFC; the cortical parcellations in the lateral parietal cortex correspond to the SPL, IPS, and inferior parietal lobule (IPL). In addition, the early visual cortex (V1) ROI was defined from the visual network located in areas labeled the striate cortex and striate calcarine. The ROIs defined by atlases are imperfect because they do not map precisely onto individual functional anatomy. Thus, the ROI analyses may include functional activations from anatomically adjacent regions. The last two visual ROIs, the FFA and PPA, were identified from an independent functional category-selective localizer. The preprocessed data of the two functional localizer runs were fitted with a first-level GLM to estimate BOLD responses to each of the stimulus categories in the individual volumetric brain. The FFA was defined by a contiguous cluster of voxels in the fusiform gyrus from the contrast between faces vs. all remaining categories, at p<0.001, uncorrected; the PPA was defined as the contiguous cluster of voxels in the parahippocampal gyrus from the contrast between scenes/corridors vs. all remaining categories, at *p*(FWE)<0.005. No activation was observed at these thresholds for six participants, thus a more lenient threshold was used to allow sufficient voxels in FFA and PPA to be identified for analyses. The average number of voxels for each ROI is reported in *Supplementary file 4*. Note that a large portion of IFJp and IFJa from the HCP-MMP1 atlas overlapped with the vLPFC from the 17-network atlas. In addition, PEF from the HCP-MMP1 atlas highly overlapped with the precentral label from the dorsal attention network in the left hemisphere and the ventral attention network in the right hemisphere using the 17-network atlas (*Figure 2—figure supplement 1*).

## Univariate whole-brain analysis

Although our primary goal was to identify cortical involvement in holding the search template during the face cue and delay periods, we also examined the scene validity effect during the search display period of the *face search task*. To do this, the normalized and smoothed data for the scene-valid and scene-invalid conditions were entered into a second-level GLM with random effects modeled at the group level. Linear contrasts between the two conditions were used to reveal selective regions with significant BOLD activations to either the scene-valid (valid>invalid) or scene-invalid (invalid>valid) conditions. The resultant *t*-value maps were corrected for multiple comparisons at the cluster level using the threshold-free cluster enhancement (TFCE; *Smith and Nichols, 2009*; *Spisák et al., 2019*) implemented in SPM12. The threshold was set at *p*TFCE<0.005 for clusters with greater than 50 voxels.

## Multivariate pattern analysis

The estimated beta parameters from the *face search task* and the *face and scene 1-back task* first-level GLMs were used to decode whether the target face and/or its associated scene was held as the search template during the *search cue* or *search delay* periods. The beta parameters extracted from each of the 12 ROIs were normalized to remove univariate differences between conditions before being submitted to a binary linear SVM classifier implemented in LIBSVM (*Chang and Lin, 2011*), with a default cost parameter of 1. *Figure 2C* illustrates the single-step cross-classification scheme (*Brandman and Peelen, 2017*; *Gayet and Peelen, 2022*) adopted in the current study. For the decoding of faces during the search cue period, the classifiers were trained to discriminate between

pairs of faces (with a full combination of six pairs based on four types of face cues) on the *face search task* runs. These classifiers were then tested using *face and scene* information from the 1-back task stimulus sample period. Following the same procedure for the delay period, classifiers were trained to discriminate between pairs of *search delay* period information following each face cue (six pairs of delay periods); the classifiers were then tested on the delay period following face or scene sample stimuli from the *face and scene 1-back task*. The classification accuracy was estimated by averaging performance from the six binary classifiers for each type of cross-classification.

## Statistical inference

Group-level statistical significance testing was established based on a nonparametric permutation method in which data labels were randomized. The null hypothesis classification accuracy distribution was generated by 10,000 iterations of each type of cross-classification for each participant. The group-level null distribution was then calculated by averaging these classification accuracies across participants. The permuted p-value was defined by the proportion of counts in the null distribution that were equal to or higher than the observed real group average classification accuracy, $(n+1)/(10,000+1)$. This p-value was used as the assessment for statistical significance, and all resulting p-values were controlled for family-wise error by using Bonferroni correction.

## Searchlight analysis

We also investigated the whole-brain multivariate decoding results using the searchlight approach (*Kriegeskorte et al., 2006*). At the individual level, the linear SVM decoding accuracy was assigned to the center voxel within a 9 mm radius sphere in volumetric space. This procedure estimates classification accuracies for all voxels within a whole-brain mask. The pairwise classification schemes were identical to those implemented for the ROI multivariate decoding analysis. The resulting individual participant whole-brain information maps for different classifications were first smoothed using a 4 mm FWHM Gaussian kernel and then normalized and tested for statistical significance against 50% chance decoding using a one-sample *t*-test (one-tailed). The statistical threshold of all group-level searchlight maps was corrected for multiple comparisons at the cluster level using TFCE.

## Acknowledgements

This research received funding supported by the National Institutes of Health under Grant R01-MH113855 to Joy J Geng, and the Humanity and Social Science Youth Foundation of the Ministry of Education in China under Grant 24XJC190010 and the Research Planning Project of the Sichuan Psychological Society Grant SCSXLXH2023001 to Zhiheng Zhou.

## Additional information

### Funding

| Funder | Grant reference number | Author |
| --- | --- | --- |
| National Institutes of Health | R01-MH113855 | Joy Geng |
| Ministry of Education of the People's Republic of China | 24XJC190010 | Zhiheng Zhou |
| Sichuan Psychological Society | SCSXLXH2023001 | Zhiheng Zhou |

The funders had no role in study design, data collection and interpretation, or the decision to submit the work for publication.

### Author contributions
Zhiheng Zhou, Conceptualization, Data curation, Formal analysis, Investigation, Visualization, Methodology, Writing – original draft, Writing – review and editing; Joy Geng, Conceptualization, Supervision, Investigation, Methodology, Writing – original draft, Writing – review and editing

### Author ORCIDs
Zhiheng Zhou ⬡ https://orcid.org/0000-0003-0687-2398
Joy Geng ⬡ http://orcid.org/0000-0001-5663-9637

### Ethics
The study adhered to the ethical principles of the Declaration of Helsinki and the NIH guidelines for ethical research. All consent forms and procedures were approved by the Institutional Review Board of the University of California, Davis (IRB number 1296375).

Reviewer #1 (Public review): https://doi.org/10.7554/eLife.104041.3.sa1
Reviewer #2 (Public review): https://doi.org/10.7554/eLife.104041.3.sa2
Reviewer #3 (Public review): https://doi.org/10.7554/eLife.104041.3.sa3
Author response https://doi.org/10.7554/eLife.104041.3.sa4

## Additional files

### Supplementary files
Supplementary file 1. Whole-brain searchlight results of brain regions showing significant decoding during the search cue period.

Supplementary file 2. Whole-brain searchlight results of brain regions showing significant decoding during the search delay period.

Supplementary file 3. Univariate contrasts related to scene validity during the search period.

Supplementary file 4. Mean (SE) number of voxels in each ROI.

MDAR checklist

### Data availability
The behavioral data, deidentified preprocessed fMRI data, analysis code, and supplemental materials are available on Open Science Framework (https://osf.io/xw8hm/).

The following dataset was generated:

| Author(s) | Year | Dataset title | Dataset URL | Database and Identifier |
| --- | --- | --- | --- | --- |
| Zhou Z, Geng JJ | 2025 | Preparatory attentional templates in prefrontal and sensory cortex encode target-associated information | https://osf.io/xw8hm/ | Open Science Framework, xw8hm |

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
