## [Editor Report · eLife Assessment]

This **important** study decoded target-associated information in prefrontal and sensory cortex during the preparatory period of a visual search task, suggesting a memory component of human subjects performing such visual attention task. The evidence supporting this claim is **compelling**, based on multivariate pattern analyses of fMRI data. The results will be of interest to psychologists and cognitive neuroscientists.

---

## [Referee Report · Reviewer #1 (Public review)]

When you search for something, you need to maintain some representation (a "template") of that target in your mind/brain. Otherwise, how would you know what you were looking for? If your phone is in a shocking pink case, you can guide your attention to pink things based on a target template that includes the attribute 'pink'. That guidance should get you to the phone pretty effectively, if it is in view. Most real-world searches are more complicated. If you are looking for the toaster, you will make use of your knowledge of where toasters can be. Thus, if you are asked to find a toaster, you might first activate a template of a kitchen or a kitchen counter. You might worry about pulling up the toaster template only after you are reasonably sure you have restricted your attention to a sensible part of the scene.

Zhou and Geng are looking for evidence of this early stage of guidance by information about the surrounding scene in a search task. They train Os to associate four faces with four places. Then, with Os in the scanner, they show one face - the target for a subsequent search. After an 8 sec delay, they show a search display where the face is placed on the associated scene 75% of the time. Thus, attending to the associated scene is a good idea. The questions of interest are "When can the experimenters decode which face Os saw from fMRI recording?" "When can the experimenters decode the associated scene?" and "Where in the brain can the experimenters see evidence of this decoding? The answer is that the face but not the scene can be read out during the face's initial presentation. The key finding is that the scene can be read out (imperfectly but above chance) during the subsequent delay when Os are looking at just a fixation point. Apparently, seeing the face conjures up the scene in the mind's eye.

This is a solid and believable result. The only issue, for me, is whether it is telling us anything specifically about search. Suppose you trained Os on the face-scene pairing but never did anything connected to search. If you presented the face, would you not see evidence of recall of the associated scene? Maybe you would see the activation of the scene in different areas and you could identify some areas as search specific. I don't think anything like that was discussed here.

You might also expect this result to be asymmetric. The idea is that the big scene gives the search information about the little face. The face should activate the larger useful scene more than the scene should activate the more incidental face, if the task was reversed. That might be true if finding is related to search where the scene context is presumed to be the useful attention guiding stimulus. You might not expect an asymmetry if Os were just learning an association.

It is clear in this study that the face and the scene have been associated and that this can be seen in the fMRI data. It is also clear that a valid scene background speeds the behavioral response in the search task. The linkage between these two results is not entirely clear but perhaps future research will shed more light.

It is also possible that I missed the clear evidence of the search-specific nature of the activation by the scene during the delay period. If so, I apologize and suggest that the point be underlined for readers like me.

Comments on revised version:

I am satisfied with the revision.

---

## [Referee Report · Reviewer #2 (Public review)]

Summary:

This work is one of the best instances of a well-controlled experiment and theoretically impactful findings within the literature on templates guiding attentional selection. I am a fan of the work that comes out of this lab and this particular manuscript is an excellent example as to why that is the case. Here, the authors use fMRI (employing MVPA) to test whether during the preparatory search period, a search template is invoked within the corresponding sensory regions, in the absence of physical stimulation. By associating faces with scenes, a strong association was created between two types of stimuli that recruit very specific neural processing regions - FFA for faces and PPA for scenes. The critical results showed that scene information that was associated with a particular cue could be decoded from PPA during the delay period. This result strongly supports invoking of a very specific attentional template.

Strengths:

There is so much to be impressed with in this report. The writing of the manuscript is incredibly clear. The experimental design is clever and innovative. The analysis is sophisticated and also innovative. The results are solid and convincing.

Weaknesses:

I only have a few weaknesses to point out.

This point is not so much of a weakness, but a further test of the hypothesis put forward by the authors. The delay period was long - 8 seconds. It would be interesting to split the delay period into the first 4seconds and the last 4seconds and run the same decoding analyses. The hypothesis here is that semantic associations take time to evolve, and it would be great to show that decoding gets stronger in the second delay period as opposed to the period right after the cue. I think it would be a stronger test of the template hypothesis.

Typo in the abstract "curing" vs "during."

It is hard to know what to do with significant results in ROIs that are not motivated by specific hypotheses. However, for Figure 3, what are explanations for ROIs that show significant differences above and beyond the direct hypotheses set out by the authors?

Following the revision, I have no further comments or concerns.

---

## [Referee Report · Reviewer #3 (Public review)]

The manuscript contains a carefully designed fMRI study, using MVPA patter analysis to investigate which high-level associate cortices contain target-related information to guide visual search. A special focus is hereby on so-called 'target-associated' information, that has previously been shown to help in guiding attention during visual search. For this purpose the author trained their participants and made them learn specific target-associations, in order to then test which brain regions may contain neural representations of those learnt associations. They found that at least some of the associations tested were encoded in prefrontal cortex during the cue and delay period.

The manuscript is very carefully prepared. As far as I can see, the statistical analyses are all sound and the results integrate well with previous findings.

I have no strong objections against the presented results and their interpretation.

The authors have addressed all my previous comments and questions in their revision of the text.

---

## [Author Response]

The following is the authors’ response to the original reviews.

**Reviewer #1 (Public review):**
When you search for something, you need to maintain some representation (a "template") of that target in your mind/brain. Otherwise, how would you know what you were looking for? If your phone is in a shocking pink case, you can guide your attention to pink things based on a target template that includes the attribute 'pink'. That guidance should get you to the phone pretty effectively if it is in view. Most real-world searches are more complicated. If you are looking for the toaster, you will make use of your knowledge of where toasters can be. Thus, if you are asked to find a toaster, you might first activate a template of a kitchen or a kitchen counter. You might worry about pulling up the toaster template only after you are reasonably sure you have restricted your attention to a sensible part of the scene.Zhou and Geng are looking for evidence of this early stage of guidance by information about the surrounding scene in a search task. They train Os to associate four faces with four places. Then, with Os in the scanner, they show one face - the target for a subsequent search. After an 8 sec delay, they show a search display where the face is placed on the associated scene 75% of the time. Thus, attending to the associated scene is a good idea. The questions of interest are "When can the experimenters decode which face Os saw from fMRI recording?" "When can the experimenters decode the associated scene?" and "Where in the brain can the experimenters see evidence of this decoding? The answer is that the face but not the scene can be read out during the face's initial presentation. The key finding is that the scene can be read out (imperfectly but above chance) during the subsequent delay when Os are looking at just a fixation point. Apparently, seeing the face conjures up the scene in the mind's eye.This is a solid and believable result. The only issue, for me, is whether it is telling us anything specifically about search. Suppose you trained Os on the face-scene pairing but never did anything connected to the search. If you presented the face, would you not see evidence of recall of the associated scene? Maybe you would see the activation of the scene in different areas and you could identify some areas as search specific. I don't think anything like that was discussed here.You might also expect this result to be asymmetric. The idea is that the big scene gives the search information about the little face. The face should activate the larger useful scene more than the scene should activate the more incidental face, if the task was reversed. That might be true if the finding is related to a search where the scene context is presumed to be the useful attention guiding stimulus. You might not expect an asymmetry if Os were just learning an association.It is clear in this study that the face and the scene have been associated and that this can be seen in the fMRI data. It is also clear that a valid scene background speeds the behavioral response in the search task. The linkage between these two results is not entirely clear but perhaps future research will shed more light.It is also possible that I missed the clear evidence of the search-specific nature of the activation by the scene during the delay period. If so, I apologize and suggest that the point be underlined for readers like me.

We have added text related to this issue, particularly in the discussion (page 19, line 6), and have also added citations of studies in humans and non-human primates showing a causal relationship between preparatory activity in prefrontal and visual cortex and visual search performance (page 6, line 16).

**Reviewer #2 (Public review):**
Summary:This work is one of the best instances of a well-controlled experiment and theoretically impactful findings within the literature on templates guiding attentional selection. I am a fan of the work that comes out of this lab and this particular manuscript is an excellent example as to why that is the case. Here, the authors use fMRI (employing MVPA) to test whether during the preparatory search period, a search template is invoked within the corresponding sensory regions, in the absence of physical stimulation. By associating faces with scenes, a strong association was created between two types of stimuli that recruit very specific neural processing regions - FFA for faces and PPA for scenes. The critical results showed that scene information that was associated with a particular cue could be decoded from PPA during the delay period. This result strongly supports the invoking of a very specific attentional template.Strengths:There is so much to be impressed with in this report. The writing of the manuscript is incredibly clear. The experimental design is clever and innovative. The analysis is sophisticated and also innovative. The results are solid and convincing.Weaknesses:I only have a few weaknesses to point out.This point is not so much of a weakness, but a further test of the hypothesis put forward by the authors. The delay period was long - 8 seconds. It would be interesting to split the delay period into the first 4seconds and the last 4seconds and run the same decoding analyses. The hypothesis here is that semantic associations take time to evolve, and it would be great to show that decoding gets stronger in the second delay period as opposed to the period right after the cue. I don't think this is necessary for publication, but I think it would be a stronger test of the template hypothesis.

We conducted the suggested analysis, and we did not find clear evidence of differences in decoding scene information between the earlier and later portions of the delay period. This may be due to insufficient power when the data are divided, individual differences in when preparatory activation is the strongest, or truly no difference in activation over the delay period. More details of this analysis can be found in the supplementary materials (page 12, line 16; Figure S1).

Type in the abstract "curing" vs "during."

Fixed.

It is hard to know what to do with significant results in ROIs that are not motivated by specific hypotheses. However, for Figure 3, what are the explanations for ROIs that show significant differences above and beyond the direct hypotheses set out by the authors?

We added reasoning for the other a priori ROIs in the introduction (page 4, line 26). There is substantial evidence suggesting that frontoparietal areas are involved in cognitive control, attentional control, and working memory. The ROIs we selected from frontal and parietal cortex are based on parcels within resting state networks defined by the s17-network atlases (Schaefer et al., 2018). The IFJ was defined by the HCP-MMP1 (Glasser et al., 2016). These regions are commonly used in studies of attention and cognitive control, and the exact ROIs selected are described in the section on “Regions of interest (ROI) definition”. While we have the strongest hypothesis for IFJ based on relatively recent work from the Desimone lab, the other ROIs in lateral frontal cortex and parietal cortex, are also well documented in similar studies, although the exact computation being done by these regions during tasks can be hard to differentiate with fMRI.\

**Reviewer #3 (Public review):**
The manuscript contains a carefully designed fMRI study, using MVPA pattern analysis to investigate which high-level associate cortices contain target-related information to guide visual search. A special focus is hereby on so-called 'target-associated' information, that has previously been shown to help in guiding attention during visual search. For this purpose the author trained their participants and made them learn specific target-associations, in order to then test which brain regions may contain neural representations of those learnt associations. They found that at least some of the associations tested were encoded in prefrontal cortex during the cue and delay period.The manuscript is very carefully prepared. As far as I can see, the statistical analyses are all sound and the results integrate well with previous findings.I have no strong objections against the presented results and their interpretation.
**Reviewer #1 (Recommendations for the authors):**
One bit of trivia. In the abstract, you should define IFJ on its first appearance in the text. You get to that a bit later.

Fixed.

**Reviewer #2 (Recommendations for the authors):**
I really don't have much to suggest, as I thought that this was a clearly written report that offered a clever paradigm and data that supported the conclusions. My only suggestion would be to split the delay period activity and test whether the strength of the template evolves over time. Even though fMRI is not the best tool for this, still you would predict stronger decoding in the second half of the delay period

Please see above for our response to the same comment.

**Reviewer #3 (Recommendations for the authors):**
I would just like to point out some minor aspects that might be worth improving before publishing this work.Abstract: While in general, the writing is clear and concise, I felt that the abstract of the manuscript was particularly hard to follow, probably because the authors at some point re-arranged individual sentences. For example, they write in line 12 about 'the preparatory period', but explain only in the following sentence that the preparatory period ensues 'before search begins'. This made it a bit hard to follow the overall logic and I think could easily be fixed.

We have addressed this comment and updated the abstract.

Also in the abstract: 'The CONTENTS of the template typically CONTAIN...' sounds weird, no? Also, 'information is used to modulate sensory processing in preparation for guiding attention during search' sounds like a very over-complicated description of attentional facilitation. I'm not convinced either whether the sequence is correct here. Is the information really used to (first) modulate sensory processing (which is a sort of definition of attention in itself) to (then) prepare the guidance of attention in visual search?

We have addressed this comment and updated the abstract.

The sentence in line 7, 'However, many behavioral studies have shown that target-associated information is used to guide attention,...' (and the following sentence) assumes that the reader is somewhat familiar with the term 'target-associations'. I'm afraid that, for a naive reader, this term may only become fully understandable once the idea is introduced a bit later when mentioning that participants of the study were trained on face-scene pairings. I think it could help to give some very short explanation of 'target-associations' already when it is first mentioned. The term 'statistically co-occurring object pairs', for example, could be of great help here.

Thank you for the suggestion. We have added it to the abstract.

page 2, line 22: 'prefrotnal'

Fixed.

page 2, line 24/25: 'information ... can SUPPLANT (?) ... information'. (That's also a somewhat unfortunate repetition of 'information')

Fixed.

page 4, line 23-25: 'Working memory representations in lateral prefrontal and parietal regions are engaged in cognitive control computations that ARE (?) task non-specific but essential to their functioning'

Fixed.

page 7, line 1: maybe a comma before 'suggesting'?

Fixed.

page 7, line 14-16: Something seems wrong with this sentence: 'The distractor face was a race-gender match, which we previously FOUND MADE (?) target discrimination difficult enough to make the scene useful for guiding attention'

We have addressed this comment and rewritten this part (now on page 7, line 18).

Results / Discussion sections:In several figures, like in Fig3A, the three different IFJ regions, are grouped separately from the other frontal areas, which makes sense given the special role IFJ plays for representing task-related templates. However, IFJ is still part of PFC. I think it would be more correct to group the other frontal areas (like FEF vLPFC etc.) as 'Other Frontal' or even 'Other PFC'.

We have made the changes based on the reviewer’s suggestion.

In some of the Figures, e.g. Fig 3 and 5, I had the impression that the activation patterns of some conditions in vLPFC were rather close to the location of IFJ, which is just a bit posterior. I think I remember that functional localisers of IFJ can actually vary quite a bit in localisation (see e.g. in the Baldauf/Desimone paper). Also, I think it has been shown in the context of other regions, like the human FEF that its position when defined by localisation tasks is not always nicely and fully congruent with the respective labels in an atlas like the Glasser atlas. It might help to take this in consideration when discussing the results, particularly since the term vLPFC is a rather vague collection of several brain parcels and not a parcel name in the Glasser atlas. Some people might even argue that vLPFC in the broad sense contains IFJ, similar to how 'Frontal' contains IFJ (see above). How strong of a point do the authors want to make about activation in IFJ versus in vlPFC?

We have now added text discussing the inability to truly differentiate between subregions of IFJ and other parts of vLPFC in the methods section on ROIs (page 25, line 13) and in the discussion (page 18, line 25). However, one might think that it is even more surprising given the likely imprecision of ROI boundaries that we see distinct patterns between the subregions of IFG defined by Glasser HCP-MMP1 and the other vLPFC regions defined by the 17-network atlases. We do not wish to overstate the precision of IFJ regions, but note the ROI results within the context of the larger literature. We are sure that our findings will have to be reinterpreted when newer methods allow for better localization of functional subregions of the vLPFC in individuals.

Given that the authors nicely explain in the introduction how important templates are in visual search, and given that FEF has such an important role in serially guiding saccades through visual search templates, I think it would be worth discussing the finding that FEF did not hold representation of these targets. Of course, this could be in part due to the specific task at hand, but it may still be interesting to note in the Discussion section that here FEF, although important for some top-down attention signals, did not keep representations of the 'search' templates. Is it because there is no spatial component to the task at hand (like proposed in Bedini 2021)?

We have now added text directly addressing this point and citing the Bedini et al. paper in the discussion (page 18, line 18). Besides our current findings, the relationship between IFJ and FEF is really interesting and will hopefully be investigated more in the future.

Page 18, line 5: 'we the(N) associated...'

Fixed.